# Genome-Wide Identified MADS-Box Genes in *Prunus campanulata* ‘Plena’ and Theirs Roles in Double-Flower Development

**DOI:** 10.3390/plants12173171

**Published:** 2023-09-04

**Authors:** Chaoren Nie, Xiaoguo Xu, Xiaoqin Zhang, Wensheng Xia, Hongbing Sun, Na Li, Zhaoquan Ding, Yingmin Lv

**Affiliations:** 1Wuhan Institute of Landscape Architecture, Wuhan 430081, China; niechaoren@bjfu.edu.cn (C.N.); breezeqin@sina.com (X.Z.); xiawensh0719@163.com (W.X.); shongbin1966@163.com (H.S.); tjdy1314tjdy@163.com (N.L.); 13995511953@139.com (Z.D.); 2Wuhan Landscape Ecology Group Co., Ltd., Wuhan 430070, China; niechaoren@163.com; 3School of Landscape Architecture, Beijing Forestry of University, Beijing 100083, China

**Keywords:** flowering cherry, MADS-box, gene family, ABCE genes, floral organ development, double flower

## Abstract

The MADS-box gene family plays key roles in flower induction, floral initiation, and floral morphogenesis in flowering plants. To understand their functions in the double-flower formation of *Prunus campanulata* ‘Plena’ (hereafter referred to as PCP), which is an excellent flowering cherry cultivar, we performed genome-wide identification of the MADS-box gene family. In this study, 71 MADS-box genes were identified and grouped into the Mα, Mβ, Mγ and MIKC subfamilies according to their structures and phylogenetic relationships. All 71 MADS-box genes were located on eight chromosomes of PCP. Analysis of the cis-acting elements in the promoter region of MADS-box genes indicated that they were associated mainly with auxin, abscisic acid, gibberellin, MeJA (methyl jasmonate), and salicylic acid responsiveness, which may be involved in floral development and differentiation. By observing the floral organ phenotype, we found that the double-flower phenotype of PCP originated from petaloid stamens. The analysis of MIKC-type MADS-box genes in PCP vegetative and floral organs by qRT–PCR revealed six upregulated genes involved in petal development and three downregulated genes participating in stamen identity. Comparative analysis of petaloid stamens and normal stamens also indicated that the expression level of the *AG* gene (*PcMADS40*) was significantly reduced. Thus, we speculated that these upregulated and downregulated genes, especially *PcMADS40*, may lead to petaloid stamen formation and thus double flowers. This study lays a theoretical foundation for MADS-box gene identification and classification and studying the molecular mechanism underlying double flowers in other ornamental plants.

## 1. Introduction

Flowering cherries, belonging to the Prunus subgenus *Cerasus* in the family Rosaceae, are very popular ornamental trees and are widely cultivated worldwide [1]. *Prunus campanulata* ‘Plena’ (PCP) is a double-flower cultivar of *P*. *campanulata* native to the Taiwan region of China [2]. Due to its elegant floral shape and brilliant colour, it is increasingly used in the construction of city gardens and beautiful villages in the Yangtze River basin and southern regions of China.

The MADS-box gene family is a key gene family distributed universally in animals, plants and fungi [3]. In plants, its members not only regulate floral organ development, flowering time determination and reproductive growth but are also involved in the development of fruits, roots, stems and leaves [4,5,6,7,8,9]. Generally, this gene family contains highly conserved MADS-box domains, which consist of the yeast transcription factor *Mcm1*, *Arabidopsis* flower homologous gene *AG*, goldfish flower homologous gene *DEF* and human serum response factor *SRF* [10,11]. By sequence structure, the MADS-box gene family can be divided into two clades: type I and type II [12]. Type I encodes one *SRF-like* MADS domain, including only one to two exons [13], whereas type II, composed of MIKC* and MIKC^c^ structures in plants, encodes *MEF2-like* and MIKC-type MADS domains. The MIKC-type MADS protein sequences consist of a semiconservative K-box (keratin-like) domain, a poorly conserved I-box (intervening) domain and a variable C-terminal domain [14,15,16]. MIKC-type gene functions are well-known to regulate the development of floral organs and fruits, flowering time and gametophytic cell division [9,16]. This type of gene is also further divided into 12 or 13 groups (one group is absent in *Arabidopsis*), namely, *AG* (class C genes), *AGL6*, *AGL12, AP3-PI* (class B genes), *Bs*, *SOC1*, *SVP*, *SEP* (class E genes), *AGL17*, *AP1-FUL* (class A genes), *AGL15*, *FLC*, and *TM8* [17].

The MIKC-type genes in plants were first identified as floral organ development genes in *A*. *thaliana* and *Antirrhinum majus*. According to gene functions, they were grouped into five classes: A, B, C, D and E. Different class combinations are considered to regulate the development of sepals (A + E), petals (A + B + E), stamens (B + C + E), carpels (C + E), and ovules (D + E) [18]. In *A*. *thaliana*, 107 MADS-box genes have been identified, and most of their functions have also been revealed [19,20].

The double-flower phenotype is a very important breeding object in ornamental plants. The origin of double flowers mainly includes six types: pistil and stamen origin, repetition origin, accumulation origin, garret origin, bract origin, and inflorescence origin [21]. Flowering cherry cultivars have several petal number configurations, mainly including single petals (petals: 5–10), semidouble petals (petals: 11–20), double petals (petals: 21–50), chrysanthemum petals (petals > 50), duplicate petals (double corollas) and so on [22]. Most double flowers in cherry cultivars, mainly belonging to *Prunus serrulata* var *lannesiana,* originate from stamen or pistil petalization [23]. Studies of the molecular mechanism underlying double flowers primarily focus on the *AG* gene. Loss of function of the *AG* gene or a decrease in its expression level causes stamens to convert into petals [24,25,26]. However, the molecular regulatory mechanism of double-flower development in PCP is unclear. In this study, we first identified MADS-box genes in *P*. *campanulata* ‘Plena’ (PCP) and then analysed their phylogenetic relationships, structure, localization on chromosomes, duplication patterns, conserved motifs, and cis-acting elements. RNA-seq and qRT–PCR analyses of vegetative and floral organs were employed to explore MADS-box gene expression patterns and ABCE gene expression levels. This study presents a genome-wide analysis of the MADS-box gene family in PCP and their roles in double-flower development. These results will also help us understand the molecular mechanism underlying double-flower development in other ornamental plants.

## 2. Results

### 2.1. MADS-Box Gene Family Identification

We used two methods to identify the MADS-box gene family in the PCP genome. First, we used Arabidopsis MADS-box protein sequences as queries to perform a BLASTP search against the PCP genome. A total of 58 candidate sequences were obtained. Then, HMMER software was employed to perform an HMM search, and 72 candidate sequences were obtained. All candidate sequences were submitted to the Pfam website to check for conserved MADS-box domains. Except for one protein sequence, the protein sequences (71 in total) all contained the conserved MADS-box domain. Therefore, they were treated as the MADS-box gene family. We assigned their names as PcMADS01~PcMADS71 (Table 1 and Appendix A). Furthermore, sequence analysis showed that the length of the MADS-box gene protein sequences varied from 136 bp to 17,349 bp. Their theoretical isoelectric points (PIs) were between 4.00 (PcMADS40) and 10.11 (PcMADS06). Their molecular weights (MWs) varied from 7096 (PcMADS16) to 45,056 (PcMADS44). Based on their conserved domains, they could be divided into two lineages (type I and type II); 27 members could be unambiguously classified as type II, and the remaining 44 members were classified as type I, as shown in Table 1.

### 2.2. Phylogenetic Analysis of the MADS-Box Gene Family

To investigate the evolution and classification of the MADS-box gene family, phylogenetic trees of type I and type II MADS-box genes based on *A*. *thaliana* and PCP protein sequences were constructed with the neighbour-joining (NJ) method, as shown in Figure 1a,b, respectively. As Figure 1a shows, type I MADS-box genes could be divided into three subfamilies: Mα, Mβ and Mγ. Subfamily Mα, with 25 members, was the largest subfamily, subfamily Mβ consisted of 10 members, and the smallest subfamily was Mγ, with nine genes. We also found that the PCP genes first grouped with their paralogous genes and then clustered with their homologues from *A*. *thaliana* within the Mα, Mβ and Mγ subfamilies. In general, the type II MADS-box genes can be classified into the MIKC* and MIKC^c^ subfamilies [27], and the MIKC^c^ subfamily can also be divided into 13 groups, i.e., 12 groups from *A*. *thaliana* (SEP, AGL6, AP1/FUL, SOC1, FLC, AGL17, AGL15, AG, AGL12, SVP, AP3/PI, BS/TT16) and one TM8 group, which plays a key role in fertilization and seed development in tomato, grape and poplar but not in *A*. *thaliana* [28]. The main functions of the SVP group are regulating the flowering time, flower development and dormancy of the plant [29]. Three genes in this group were identified, which is fewer than that in *P. mume* [30]. The AGL17 group encodes a protein involved in promoting flowering [31], and three members were found in PCP, one more than in *P. mume*. In the SOC1 group, three genes were identified, the same number as in *P. mume*. The MICK* subfamily and ABs/TT16 group were absent in PCP.

### 2.3. Chromosome Localization and Synteny Analysis of the MADS-Box Gene Family

According to the PCP genome annotation data, all 71 MADS-box genes were located on eight chromosomes of PCP (Figure 2). The MADS-box genes were unevenly distributed on the chromosomes. Chromosome 1 (LG01) had the most MADS-box genes, up to 18, and chromosome 4 (LG04) had the fewest, with only four genes. This may be caused by uneven duplication events of chromosome fragments. The subfamilies of the MADS-box genes also had different distributions on the chromosomes. The Mα subfamily was mainly distributed on chromosomes 1 (LG01), 3 (LG03), and 4 (LG04); the Mβ subfamily was mainly distributed on chromosome 5 (LG05); the Mγ subfamily was mainly distributed on chromosomes 1 (LG01) and 6 (LG06); and the MICK subfamily was mainly distributed on chromosomes 1 (LG01) and 2 (LG02).

Synteny analysis was performed between PCP and *P. × yedoensis* ‘Somei-Yoshino’ and between PCP and *A*. *thaliana* to investigate the evolutionary relationship of the MADS-box genes (Figure 3a,b). The numbers of collinear genes and gene pairs were 23,154 and 24, respectively, for the MADS-box gene family between PCP and A. thaliana and 18,737 and 36 between PCP and *P. × yedoensis* ‘Somei-Yoshino’. We also analysed gene duplication in the MADS-box gene family of PCP and *A*. *thaliana* by MCScanX tool in TBtools v1.096, looking for WGD or segmental duplication, eight tandem duplications and 39 dispersed genes. In *A*. *thaliana*, there were 50, 11, and 24 corresponding genes. The results revealed that the expansion mechanisms were different in PCP compared to *A*. *thaliana*. For *A*. *thaliana*, WGD or segmental events played a more important role in MADS-box gene family expansion, whereas in PCP, transposition events and transposition events were the main mechanisms.

### 2.4. MADS-Box Gene Structure and Conserved Motif Analysis

The MADS-box gene structure and conserved motif analysis were performed by the MEME program, and the results are shown in Figure 4A,B. Among the top 10 motifs, motif 1 was the MADS domain, and motif 2 denoted the K domain. All MADS-box genes contained motif 1 except for PcMADS03, 28, 29, 42, 45, and 25. However, the InterPro analysis suggested that the MADS domains actually existed in these genes. Motifs 8 and 10 were mainly distributed in the Mα subfamily, and motif 10 was unique to the Mα subfamily. Motif 7 was specific to the Mβ subfamily. The Mγ subfamily contained motifs 1, 4, and 5, and motifs 4 and 5 were unique to this subfamily. Motifs 1, 2 and 3 were in the MIKC subfamily, and motif 2, as the K-box, was unique to this subfamily. We found that the exon numbers per gene varied from one to 12. The average exon number of type II (7.59) was greater than that of type I (1.36).

### 2.5. Prediction of Cis-Acting Elements of the Promoter Region

To identify cis-acting elements, we analysed the 2000 bp upstream promoter regions of 71 MADS-box genes in PCP. A total of 105 cis-acting elements were predicted and annotated. All of these gene promoter regions had cis-acting elements, and their distributions are shown in Figure 5. The number of cis-acting elements per gene varied from nine to 70. The three genes PcMADS10, PcMADS30 and PcMADS03 had the most cis-acting elements, containing 70, 44 and 42, respectively. Furthermore, these cis-acting elements with the highest frequency were involved in auxin, abscisic acid, gibberellin, MeJA (methyl jasmonate), and salicylic acid responses. These cis-acting elements related to hormones may be directly or indirectly involved in floral organ development.

### 2.6. Phenotypic Identification of Double Flowers in PCP

*P. campanulata*, which is a wild species with single flowers, has four whorls of normal floral organs, including five sepals in whorl 1, five petals in whorl 2, 30 to 49 stamens (on average 40.67 ± 5.63) in whorl 3, and one carpel in whorl 4, as shown in the right photo of Figure 6a. In contrast, PCP, as a cultivar of *P. campanulata*, is double-flowered, with five normal sepals in whorl 1, 23 to 32 petals (on average 28.33 ± 3.13) in whorl 2, 16 to 27 malformed or normal stamens (on average 20.40 ± 4.15) in whorl 3, and one carpel in whorl 4, as shown in the left photo of Figure 6a. The stamen number of PCP was lower than that of *P. campanulata*. Many malformed anthers could not produce pollen, and the number of pollen grains in normal anthers was also less than that in *P. campanulata*. In addition, there was also consistency in the sum of the second- and third-whorl floral organs. The average of the sum of the second- and third-whorl floral organs of *P. campanulata* was 45.67 ± 5.14, while that in PCP was 47.07 ± 3.47, with no statistically significant difference (*p* < 0.05). Figure 6b shows the different degrees of stamen petalization in the flowers. Rows Ⅰ and Ⅴ are the normal petals and stamens of PCP, respectively. Rows II to IV show stamens with different degrees of petalization. Compared with normal petals, these petals were slightly smaller, and the base had a petiole. Thus, we believed that the double flowers of PCP originated from the petaloid stamens.

### 2.7. MIKCc-Type MADS-Box Gene Expression Analysis

To understand the expression patterns of the MIKC-type MADS-box genes in different organs and ascertain their roles in floral organ development, we carried out RNA-seq of four vegetative organs and structures (root, stem, leaf and petiole) and five floral structures (sepal, petal, stamen, petaloid stamen and pistil), and the results are shown in Figure 7. The figure shows that some groups were highly conserved and that the expression patterns were similar. For example, SOC1 and SVP were mainly expressed in vegetative organs and structures. However, there were different expression patterns in the same groups. For example, PcMADS05 and PcMADS68, belonging to the AP1/FUL group, were mainly expressed in sepals and carpels, while PcMADS46, the other member of this group, was expressed in leaves and petioles. In the AGL6 group, more important differences in gene expression patterns within groups could be found. PcMADS20 was mainly expressed in sepals, petaloid stamens and pistils, while PcMADS27, another member of the AGL6 group, was expressed in four vegetative organs/structures (roots, stems, leaves, and petioles).

The results of cluster analysis of MIKC−type MADS-box gene expression patterns are also shown in Figure 7b. In this figure, two clusters that corresponded to genes preferentially expressed in vegetative and floral organs were observed. The first cluster included two expression groups corresponding to the roots and other vegetative organs/structures. The first expression group included four genes that were expressed mainly in the roots. Two (PcMADS14 and PcMADS34) were members of the AGL17 group, and the other two were PcMADS15 and PcMADS17, belonging to the AGL12 and TM8 groups, respectively. These genes were highly expressed in roots, although they were also expressed in other vegetative organs/structures. The second expression group (10 genes) included all three members of the SOC1 and SVP groups and genes from the AGL6 (PcMADS27), AGL17 (PcMADS64), FLC (PcMADS50), and AP1/FUL (PcMADS46) groups. They were mainly expressed in three vegetative organs/structures: stems, leaves and petioles. The second cluster included four major expression groups. The first expression group contained five genes highly expressed in the pistils, except for PcMADS04, belonging to the SEP group, which was expressed in the stems. PcMADS05 (AP1/FUL) was also highly expressed in the sepals, and the other genes were PcMADS40 (AG), PcMADS68 (AP1/FUL), and PcMADS69 (SEP). The second expression group contained two genes, PcMADS25 and PcMADS59 (AGL15), which were highly expressed in the stamens. The third group contained two genes, PcMADS46 and PcMADS10 (AP3/PI), which were mainly expressed in the petals, petaloid stamens and stamens. They were class B genes and were involved in the petal development process. The fourth expression group included four members from the SEP (PcMADS47 and PcMADS03), AGL6 (PcMADS20), and AP3/PI (PcMADS06) groups. PcMADS47 was mainly expressed in the sepals, petals and pistils, while the other three were expressed in the petals, petaloid stamens and pistils [5].

Based on the above RNA-seq results, we analysed the gene expression levels between stamens and petaloid stamens. Compared with normal stamens, in petaloid stamens, the expression levels of *PcMADS20* (*AGL6*), *PcMADS06* (*AP3/PI*), *PcMADS56* (*AP3/PI*), *PcMADS10* (*AP3/PI*), *PcMADS47* (*SEP*), and *PcMADS03* (*SEP*) increased. However, the three genes *PcMADS40* (*AG*), *PcMADS25* (*AGL15*), and *PcMADS59* (*AGL15*) showed decreased expression levels. The differentially expressed genes may be related to the development of petaloid stamens.

### 2.8. Expression Analysis of ABCE Class Genes in PCP by qRT–PCR

The results of the expression analysis of PCP ABCE class genes in five floral organs by qRT–PCR are shown in Figure 8. Three AP3/FUL genes were highly expressed in sepals, while they were expressed at low levels in the other four floral organs/structures. The expression patterns of the three AP3/PI genes were different. Those of PcMADS06 and PcMADS10 were similar, with the highest expression levels in sepals. PcMADS56 had the highest expression level in petaloid stamens, followed by petals and stamens, with the lowest expression in pistils. PcMADS40 belongs to the class C AG genes, with the highest expression level in sepals, followed by stamens and pistils, and the lowest expression levels in petaloid stamens and petals. This meant that the expression level of this gene in petaloid stamens was lower than that in stamens. The expression level change of these two genes in petaloid stamens may indicate that they are involved in regulating stamen petalization. The expression patterns of PcMADS47 and PcMADS03 were relatively similar, with the highest expression level in the sepals and lower expression levels in the other four parts. However, PcMADS69 and PcMADS04 had lower expression levels in the other four parts except for the sepals.

## 3. Discussion

The MADS-box gene family is one of the most important gene families in flowering plants and is involved in flower induction, floral initiation, and floral morphogenesis [28]. In this study, we identified 71 MADS-box genes in the PCP genome. According to the conserved domains, we classified the MADS-box genes into five subfamilies: Mα, Mβ, Mγ, MIKC^c^ and MICK*. Surprisingly, the MICK* subfamily was absent. However, there were five members in Mei flowers, four in peaches, 17 in apples, 10 in strawberries, five in pears, and seven in *A*. *thaliana* (Appendix A) [30,32,33,34,35]. In *A*. *thaliana*, the MIKC* subfamily is the main regulator of transcriptome dynamics during male reproductive cell development and regulates a transcription switch that directs pollen maturation [27]. The PCP stamen development was abnormal, with petaloid anthers and sterile pollen grains, which meant that the male floral organs and male gamete development were not normal. This phenomenon may be caused by the loss of the MIKC* subfamily genes.

Generally, the number of MADS box genes is related to the genome size and phylogenetic status of the species. Basal species contain fewer MADS-box genes than more derived species, which seems to be determined by whole-genome duplication (WGD) [36,37]. Appendix A shows that the total number of MADS-box genes in apple (*Malus × domestica*) and pear (*Pyrus bretschneideri*) was significantly higher than that in the other four species in the Rosaceae family. The WGD events that occurred in the two species caused the genome to become larger and the number of chromosomes to increase. Thus, the numbers of MADS-box genes were similar, ranging from 71 to 83, which are also higher than the numbers of MADS-box genes in grape (*Vitis vinifera*) and lotus (*Nelumbo nucifera*).

According to the ABCE floral organ identification model, the tetrameric protein complexes of MIKC-type genes in classes A, B, C and E identified the four whorls of floral organs: sepals, petals, stamens, and pistils [36]. Therefore, we focused on the expression patterns of ABCE class genes in different floral organs of PCP (Figure 7a). Many studies have shown that AG homologue downregulation could lead to petaloid stamens in the double-flower phenotype of some plants [18,24,25,26,38,39,40,41]. In our study, we detected nine differentially expressed genes by RNA-seq. The AG gene PcMADS40 and AGL15 genes PcMADS25 and PcMADS59 were downregulated in petaloid stamens compared to stamens. Similarly, the downregulated expression of the AG gene PrseAG resulted in double-flower formation in *Prunus serrulata* var. *lannesiana* [23]. Ectopic expression of the EjAG gene in double-flower loquat (*Eriobotrya japonica*) rescued the development of stamens and carpels from the double-flower phenotype in an Arabidopsis ag mutant [41]. In addition, we also found that six genes in the AGL6 group (PcMADS20) and AP3/PI group (PcMADS06, PcMADS56 and PcMADS10), and SEP group (PcMADS47 and PcMADS03) were upregulated in petaloid stamens vs. stamens. These results suggested that these six genes were involved in the transformation of stamens to petals. AGL6 is considered to be similar to AP1, which is highly expressed in tepals and involved in determining petals [38,42,43]. Some studies have also shown that PI has a high expression level in petaloid stamens, forming a tetramer together with AP1, AP3 and SEP to regulate the formation of petaloid stamens [44,45]. In our study, AGL6 showed an expression pattern similar to those of AP1, AP3/PI, and SEP, and six genes were upregulated in petaloid stamens vs. stamens, indicating that they may participate in petals determining tetramer formation and regulate the formation of petaloid stamens.

We found that the double-flower phenotype of PCP originated from petaloid stamens. The analysis of MIKC-type MADS-box genes in PCP vegetative and floral organs by qRT–PCR revealed six upregulated genes involved in petal development and three downregulated genes participating in stamen identity. Comparative analysis of petaloid stamens and normal stamens also indicated that the expression level of the AG gene (Class C) PcMADS40 among the ABCE class genes was significantly reduced. Thus, we speculated that these upregulated and downregulated genes, especially PcMADS40, may lead to stamen petalization and the formation of double flowers.

## 4. Materials and Methods

### 4.1. MADS-Box Gene Family Identification

We downloaded the PCP genome data from the website http://tree-bio.hzau.edu.cn/download/PCP/, accessed on 1 Fabruary 2022 [46]. Two strategies were used to identify the members of the MADS-box gene family in PCP. First, BLASTP software was employed to search globally for MADS-box genes, and the *A. thaliana* MADS-box protein sequences downloaded from the TAIR website (https://www.arabidopsis.org/, accessed on 27 July 2023) were used as query sequences [47]. Second, the HMM profiles (access numbers: PF01486 and PF00319) downloaded from the Pfam database (https://www.ebi.ac.uk/interpro/, accessed on 27 July 2023) were used to search against the PCP genome data using HMMER v3.3.2 with default parameters. Finally, the sequences obtained by the two methods were submitted to InterPro (http://www.ebi.ac.uk/interpro/, accessed on 27 July 2023) to check for conserved MADS domains [48]. These sequences were submitted to Prot Param (http://web.expasy.org/protparam/, accessed on 27 July 2023) for molecular weight (MW), theoretical PI (PI), and amino acid number computation.

### 4.2. Phylogenetic Analysis of the MADS-Box Gene Family

Multiple sequence alignments were carried out by the MUSCLE program in MEGA6 software with default parameters [49]. The phylogenetic tree based on the neighbour-joining (NJ) method was generated by MEGA6 with a P-distance model and 1000 bootstraps. Visualization was completed by iTOLv6.6 (https://itol.embl.de/, accessed on 27 July 2023) [50].

### 4.3. Chromosome Location and Synteny Analysis of the MADS-Box Gene family

The MADS-box gene family was located on chromosomes using Gene Location Visualize in TBtools v1.096 [51]. The one-step MCScanX tool was used to identify syntenic gene pairs in the MADS-box gene family, and the results were visualized by a dual synteny plot in TBtools.

### 4.4. MADS-Box Gene Structure and Motif Analysis

The MADS-box gene motifs and structures were predicted by MEME v5.0.5 (https://meme-suite.org/meme/, accessed on 20 Fabruary 2022), and the parameters were as follows: number of motifs, 10; minimum motif width, 6 bp; and maximum motif width, 60 bp [52]. SMART (http://smart.embl-heidelberg.de/, accessed on 22 Fabruary 2022,) and Pfam were used to annotate the motifs. Gene structure information was obtained from the PCP genome data [53]. Visualization was completed by Gene Structure View in TBtools.

### 4.5. Prediction of Cis-Regulatory Elements in the Promoter Region

TBtools was used to extract the upstream 2000 bp sequences of the MADS-box genes from the whole PCP genome and then submitted to Plant CARE (http://bioinformatics.psb.ugent.be/webtools/plantcare/html/, accessed on 3 December 2022) for predicting the cis-acting elements and annotating their functions, which were then visualized using the Simple Bio Sequence Viewer function of TBtools [51,54].

### 4.6. MADS-Box Gene Family Expression Analysis

To investigate the expression patterns of the PCP MADS-box genes, transcriptome sequencing (RNA-seq) was performed. Root, stem, leaf, and petiole samples of PCP were collected at the Institute of Landscape Architecture in Wuhan (Wuhan, China). Three biological replicates were used per sample. Total RNA was extracted by RNAiso Plus from TaKaRa (TAKARA Biotech, Beijing, China), and the Reagent Kit with gDNA Eraser was used for reverse transcription, both of which were performed according to the instructions. RNA-seq was performed by the Wuhan Hope Group Biology Co., Ltd. (Software New Town, Huacheng Avenue, Wuhan, China).

### 4.7. ABCE Class Gene Expression Analysis in PCP by qRT–PCR

To understand the role of ABCE genes in PCP double-petal flowers, we conducted qRT–PCR analysis. The RNA extraction and quality inspection methods were the same as those described in Section 4.6. The primers were designed according to the CDSs by Primer 5.0 software. All primers were verified by semiquantitative RT–PCR, and the list of primers used is shown in Appendix A. Primers were produced by Shanghai Biotechnology Co., Ltd. The ACTIN gene was used as the internal reference. qRT–PCR was carried out on the Roche Light Cycler 480 Ⅱ platform. Data analysis was performed with Excel, graphs were drawn by R 4.2.2 software, and significance was determined with ANOVA in R 4.2.2 software; *p* < 0.05 was considered significant. Relative gene expression was calculated by the 2^−ΔΔCt^ method [54,55].

## Figures and Tables

**Figure 1 plants-12-03171-f001:**
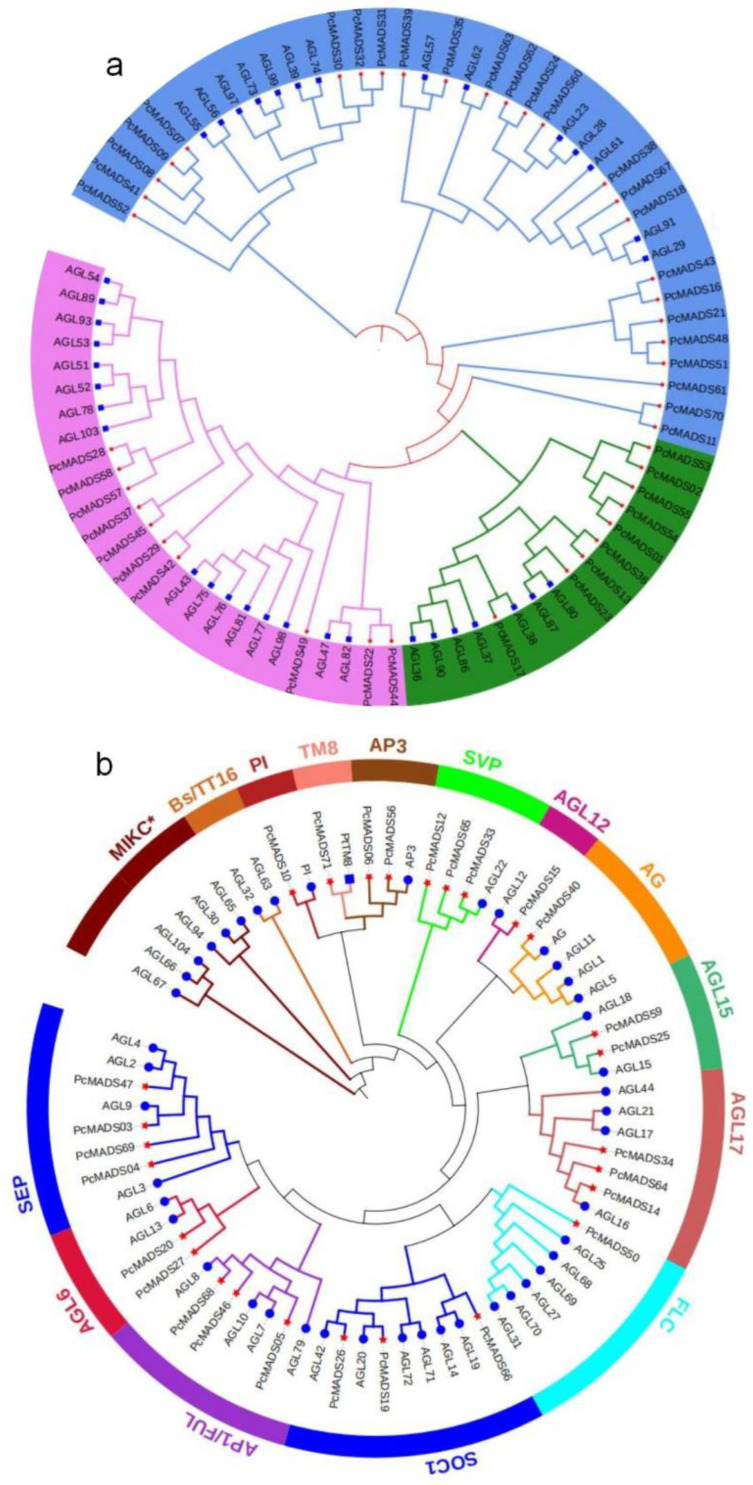
Phylogenetic trees of type I and type II MADS-box genes. (**a**) The phylogenetic tree of Type I based on the *P. campanulata* ‘Plena’ (PCP) and *A. thaliana* protein sequences reconstructed with the neighbour-joining (NJ) method. The blue branch denotes the Mα subfamily, the red branch denotes the Mβ subfamily, and the green branch denotes the Mγ subfamily. The red star at the end of the branch denotes PCP, and the blue square denotes *A. thaliana*. (**b**) The phylogenetic tree of Type II based on PCP and *A. thaliana* protein sequences reconstructed with the neighbour-joining (NJ) method. The red star at the end of the branch denotes PCP, and the blue circle denotes *A. thaliana*.

**Figure 2 plants-12-03171-f002:**
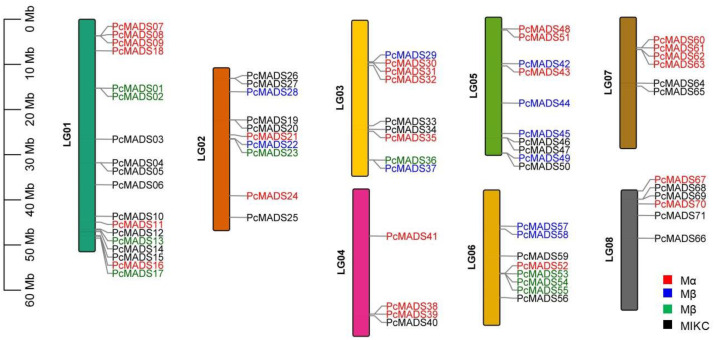
Distribution of the MADS-box genes on the eight chromosomes of *P. campanulata* ‘Plena’ (PCP). The black lines on the right side of the chromosomes denote the location of each MADS-box gene; red gene labels denote the Mα subfamily, blue gene labels denote the Mβ subfamily, green gene labels denote the Mγ subfamily, and black gene labels denote the MIKC subfamily.

**Figure 3 plants-12-03171-f003:**
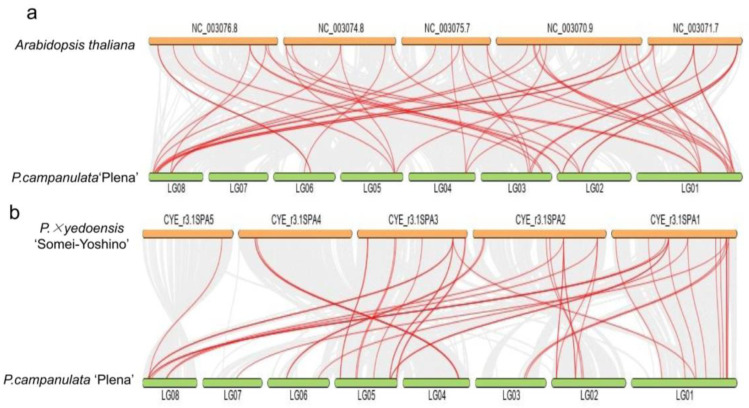
Synteny analysis of MADS-box genes between *P. campanulata* ‘Plena’ (PCP) and *A*. *thaliana* (**a**) and PCP and *P. × yedoensis* ‘*Somei-Yoshino*’ (**b**). The collinear blocks between PCP and the two plant genomes are shown as the gray lines in the background, while the syntenic MADS-box gene pairs are highlighted with red lines.

**Figure 4 plants-12-03171-f004:**
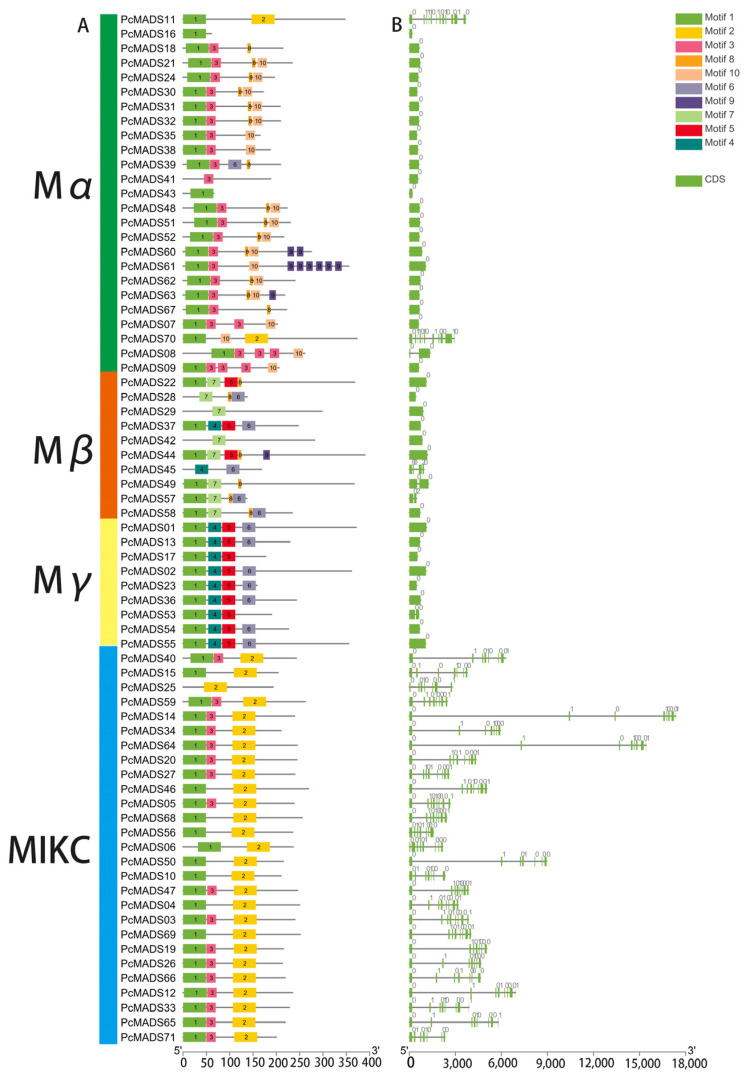
Gene structure and conserved motif analysis of *P. campanulata* ‘Plena’ (PCP) MADS-box genes. (**A**) MADS-box gene motif analysis; the number in the coloured boxes located at the mid-bottom denotes the motifs. The box length corresponds to the motif length. (**B**) MADS-box gene structure analysis. The mid-bottom blue box denotes CDSs, and the solid line denotes introns.

**Figure 5 plants-12-03171-f005:**
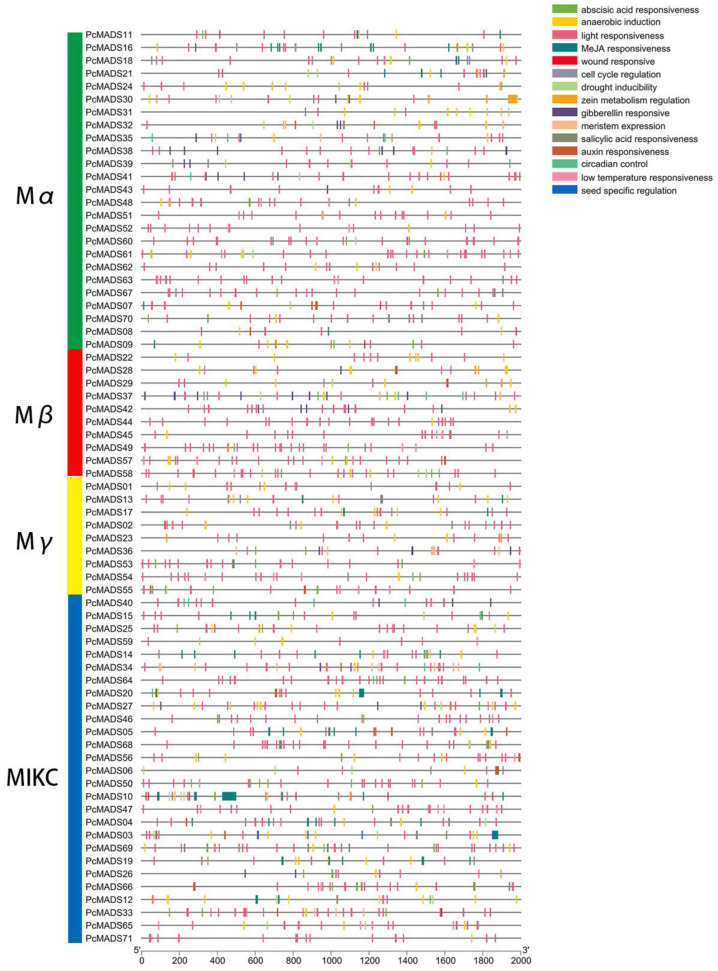
Analysis of cis-acting elements in the promoter region of MADS-box genes in *P. campanulata* ‘Plena’. These genes are shown on the left. The scale bars at the base indicate the length of the promoter sequence.

**Figure 6 plants-12-03171-f006:**
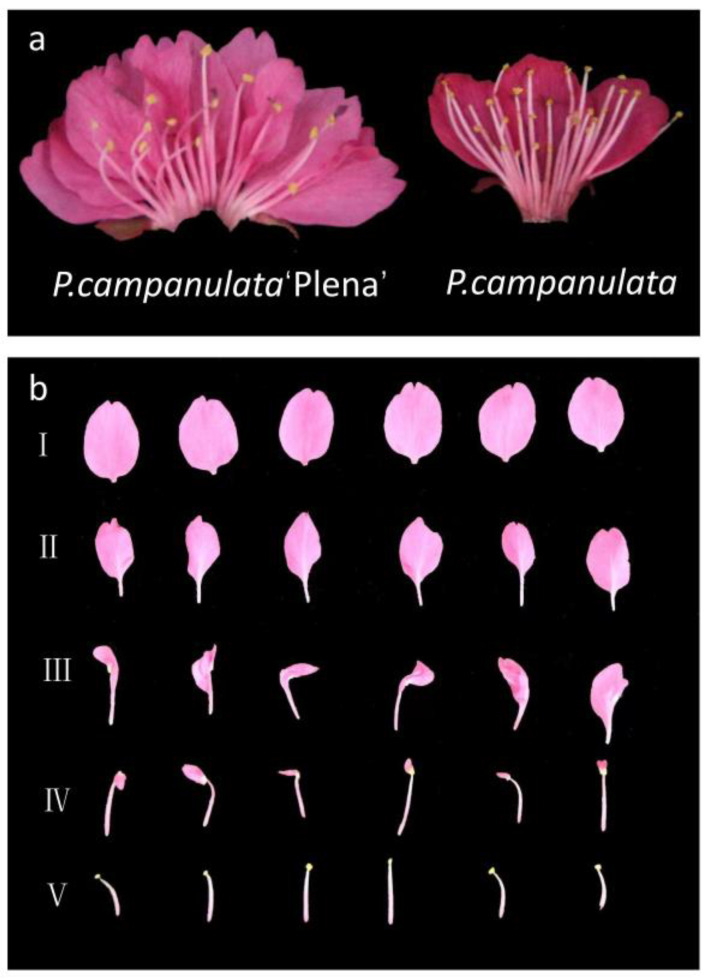
Floral phenotype of single and double flowers. (**a**) Floral phenotype photographs of *P. campanulata* ‘Plena’ (PCP) and *P. campanulata*. (**b**) Rows I~V denote stamens with different degrees of petalization in PCP flowers.

**Figure 7 plants-12-03171-f007:**
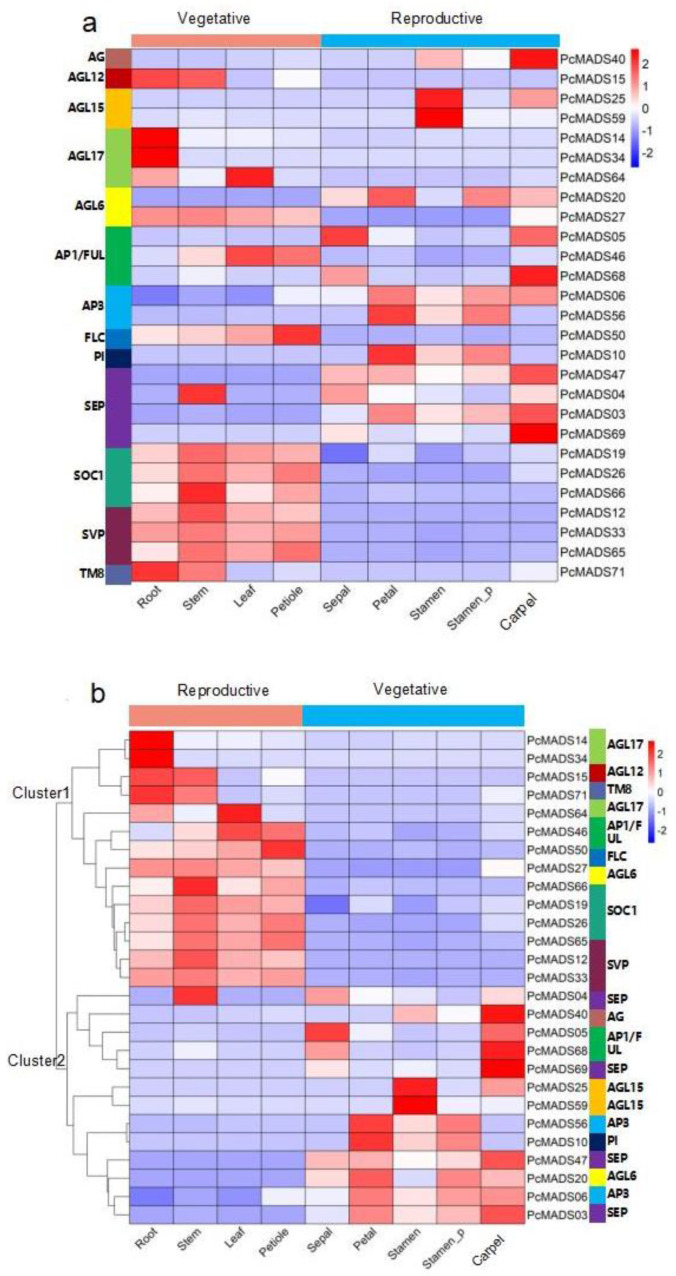
Expression analysis of *P. campanulata* ‘Plena’ MIKC−type genes in vegetative and reproductive organs by RNA−seq. Stamen_P denotes petaloid stamens. (**a**) Expression pattern according to the phylogenetic relationships. The corresponding groups of genes are annotated on the left. (**b**) Cluster analysis of the gene expression patterns. The corresponding groups of genes are indicated on the right. Colour scales, representing expression level, are shown on the right.

**Figure 8 plants-12-03171-f008:**
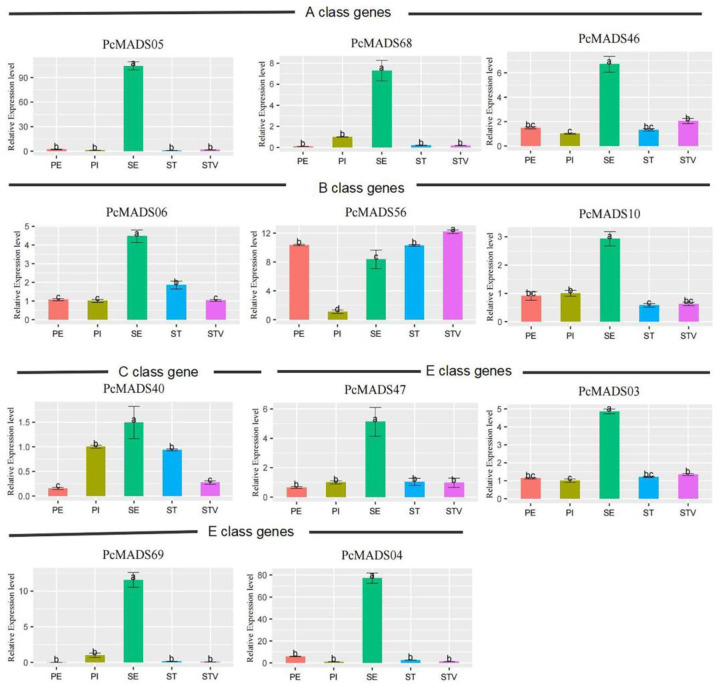
Expression analysis of *P. campanulata* ‘Plena’ ABCE class genes in floral organs by qRT–PCR. PE, PI, SE, ST, and STV denote petals, pistils, sepals, stamens, and petaloid stamens, respectively. Error bars indicate the standard deviation (SD), and different letters (a–c) indicate significant differences at *p* < 0.05.

**Table 1 plants-12-03171-t001:** The MADS-box gene family identified in PCP.

Gene Name	Gene Id	CDS (bp)	Exon Count	Protein Length (aa)	MW	PI	Type	Subfamily
PcMADS01	evm.TU.LG01.1677	1116	1	371	40,923	6.43	Type I	Mγ
PcMADS02	evm.TU.LG01.1680	1086	1	361	39,386	4.91	Type I	Mγ
PcMADS03	evm.TU.LG01.2444	3849	8	240	27,570	8.79	Type II	MIKC
PcMADS04	evm.TU.LG01.3136	3183	8	250	28,746	8.24	Type II	MIKC
PcMADS05	evm.TU.LG01.3137	2657	8	238	27,871	8.45	Type II	MIKC
PcMADS06	evm.TU.LG01.3902	2179	8	236	27,631	10.11	Type II	MIKC
PcMADS07	evm.TU.LG01.462	609	1	202	21,822	4.74	Type I	Mα
PcMADS08	evm.TU.LG01.464	1345	2	261	28,905	7.73	Type I	Mα
PcMADS09	evm.TU.LG01.465	621	1	206	22,364	4.16	Type I	Mα
PcMADS10	evm.TU.LG01.4975	2325	7	210	24,310	8.4	Type II	MIKC
PcMADS11	evm.TU.LG01.5169	3670	12	347	39,188	5.31	Type I	Mα
PcMADS12	evm.TU.LG01.5349	6915	8	235	26,615	5.31	Type II	MIKC
PcMADS13	evm.TU.LG01.5368	690	1	229	26,436	9.28	Type I	Mγ
PcMADS14	evm.TU.LG01.5407	17,349	8	239	27,368	9.47	Type II	MIKC
PcMADS15	evm.TU.LG01.5499	3775	7	204	23,253	7.74	Type II	MIKC
PcMADS16	evm.TU.LG01.5511	186	1	61	7096	9.93	Type I	Mα
PcMADS17	evm.TU.LG01.5564	534	1	177	20,236	6.76	Type I	Mγ
PcMADS18	evm.TU.LG01.785	645	1	214	23,580	9.08	Type I	Mα
PcMADS19	evm.TU.LG02.1730	5045	7	215	24,830	9.24	Type II	MIKC
PcMADS20	evm.TU.LG02.1732	4340	8	244	27,946	9.2	Type II	MIKC
PcMADS21	evm.TU.LG02.2071	705	1	234	26,130	8.97	Type I	Mα
PcMADS22	evm.TU.LG02.2165	1107	1	368	42,473	8.15	Type I	Mβ
PcMADS23	evm.TU.LG02.2176	480	1	159	18,298	9.1	Type I	Mγ
PcMADS24	evm.TU.LG02.2988	591	1	196	22,129	9.07	Type I	Mα
PcMADS25	evm.TU.LG02.3371	2785	7	193	22,092	5.1	Type II	MIKC
PcMADS26	evm.TU.LG02.426	4661	7	213	24,431	8.36	Type II	MIKC
PcMADS27	evm.TU.LG02.430	2588	8	240	27,638	6.86	Type II	MIKC
PcMADS28	evm.TU.LG02.905	417	1	138	16,228	9.15	Type I	Mβ
PcMADS29	evm.TU.LG03.1170	897	1	298	33,021	4.64	Type I	Mβ
PcMADS30	evm.TU.LG03.1188	519	1	172	19,651	4.54	Type I	Mα
PcMADS31	evm.TU.LG03.1189	627	1	208	23,207	4.54	Type I	Mα
PcMADS32	evm.TU.LG03.1235	630	1	209	23,355	4.56	Type I	Mα
PcMADS33	evm.TU.LG03.2185	3887	8	228	25,828	6.26	Type II	MIKC
PcMADS34	evm.TU.LG03.2269	5928	6	210	24,224	9.52	Type II	MIKC
PcMADS35	evm.TU.LG03.2316	498	1	165	18,708	9.43	Type I	Mα
PcMADS36	evm.TU.LG03.3240	732	1	243	27,584	9.32	Type I	Mγ
PcMADS37	evm.TU.LG03.3242	744	1	247	28,263	9.52	Type I	Mβ
PcMADS38	evm.TU.LG04.2508	564	1	187	20,858	9.47	Type I	Mα
PcMADS39	evm.TU.LG04.2509	630	1	209	23,872	9.16	Type I	Mα
PcMADS40	evm.TU.LG04.2573	6284	8	243	27,992	4.00	Type II	MIKC
PcMADS41	evm.TU.LG04.594	567	1	188	21,250	5.52	Type I	Mα
PcMADS42	evm.TU.LG05.1117	849	1	282	31,473	9.56	Type I	Mβ
PcMADS43	evm.TU.LG05.1150	201	1	66	7606	9.56	Type I	Mα
PcMADS44	evm.TU.LG05.1670	1173	1	390	45,056	8.61	Type I	Mβ
PcMADS45	evm.TU.LG05.2530	950	4	168	19,381	6.51	Type I	Mβ
PcMADS46	evm.TU.LG05.2661	5046	8	269	30,703	8.45	Type II	MIKC
PcMADS47	evm.TU.LG05.2662	3855	8	246	28,156	9.08	Type II	MIKC
PcMADS48	evm.TU.LG05.313	672	1	223	25,373	8.76	Type I	Mα
PcMADS49	evm.TU.LG05.3202	1252	2	367	42,008	5.92	Type I	Mβ
PcMADS50	evm.TU.LG05.3221	8940	7	215	24,330	8.82	Type II	MIKC
PcMADS51	evm.TU.LG05.347	693	1	230	25,864	9.34	Type I	Mα
PcMADS52	evm.TU.LG06.1294	651	1	216	23,904	9.32	Type I	Mα
PcMADS53	evm.TU.LG06.1306	635	2	190	21,197	8.85	Type I	Mγ
PcMADS54	evm.TU.LG06.1307	681	1	226	26,016	8.90	Type I	Mγ
PcMADS55	evm.TU.LG06.1308	1068	1	355	39,738	8.78	Type I	Mγ
PcMADS56	evm.TU.LG06.2017	1572	7	235	27,154	9.34	Type II	MIKC
PcMADS57	evm.TU.LG06.458	465	2	137	16,243	7.64	Type I	Mβ
PcMADS58	evm.TU.LG06.495	705	1	234	27,455	8.29	Type I	Mβ
PcMADS59	evm.TU.LG06.907	2469	8	262	29,850	9.34	Type II	MIKC
PcMADS60	evm.TU.LG07.1086	828	1	275	30,057	6.32	Type I	Mα
PcMADS61	evm.TU.LG07.1087	1068	1	355	38,629	5.53	Type I	Mα
PcMADS62	evm.TU.LG07.1144	723	1	240	26,943	9.06	Type I	Mα
PcMADS63	evm.TU.LG07.1145	657	1	218	24,761	9.46	Type I	Mα
PcMADS64	evm.TU.LG07.1998	15,428	8	245	28,108	9.52	Type II	MIKC
PcMADS65	evm.TU.LG07.2059	5798	8	219	24,627	6.93	Type II	MIKC
PcMADS66	evm.TU.LG08.1574	4634	7	218	25,041	9.04	Type II	MIKC
PcMADS67	evm.TU.LG08.27	669	1	222	25,391	9.15	Type I	Mα
PcMADS68	evm.TU.LG08.341	2436	8	255	29,194	8.64	Type II	MIKC
PcMADS69	evm.TU.LG08.342	4013	8	251	28,613	8.82	Type II	MIKC
PcMADS70	evm.TU.LG08.479	2939	11	373	41,912	5.77	Type I	Mα
PcMADS71	evm.TU.LG08.917	2323	7	200	23,175	9.9	Type II	MIKC

## Data Availability

A total of 27 raw transcriptome datasets (9 tissues with triplicates) produced by the Illumina NovaSeq 6000 platform were obtained from the website http://tree-bio.hzau.edu.cn/download, accessed on 1 Fabruary 2022.

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
