# Peer review of "Genome-Wide Identified MADS-Box Genes in Prunus campanulata ‘Plena’ and Theirs Roles in Double-Flower Development"

_plants, 2023, doi:10.3390/plants12173171_

Round 1
Reviewer 1 Report
Dear Authors
The manuscript's readability is hindered by the grammar and language. The Figures are of good representation. Unless, the readability is improved, it is difficult to review the manuscript.
Dear Authors
The manuscript needs extensive editing for English language.
Reviewer 2 Report
The authors explored the MADS gene family of PCP species and analyzed its bioinformatics and expression patterns. The results laid a foundation for further study of flower organ development in X species. But there are some issues that need to be revised
1. Some language errors require full revision, such as line 15, which is is a very excellent;
2. Why is ABCE, lacked D gene? Species PCP is missing the D gene?
3. There are only 71 MADS genes in PCP, and the number of genes is indeed relatively small. It is necessary to add a discussion on the number of MADS genes of different species in the discussion section
4. Figure 2 and 5, The font in the picture is too small to read.
The format of the references is not standardized
Reviewer 3 Report
The research delves into the function of MADS-box genes within the double-flowered cherry cultivar, Prunus campanulata ‘Plena’ (PCP). The study identifies and classifies 71 MADS-box genes into different subfamilies. These genes play a crucial role in the development and differentiation of flowers. The investigation scrutinizes their expression patterns in relation to the double-flower characteristic. Up-regulated genes are linked with petal growth, while down-regulated genes relate to stamen formation. Specifically, PcMADS40, classified as a Class C gene, displays diminished expression in petalated stamens, indicating its potential involvement in the process of double-flower development. This study establishes the groundwork for comprehending MADS-box genes and their impact on the formation of double-flowers. The study's objectives are clear, and its analysis holds credibility. While no major concerns are raised, there are some minor suggestions to consider:
Regarding Table 1, which appears somewhat extensive and comprises a list of the MADS-box gene family, it might be beneficial to condense the pertinent information into a new table, transferring the existing content to a supplementary table.
As for Figure 1, a suggestion is made to ensure consistency in the representation of parts "a" and "b" by redrawing them.
Round 2
Reviewer 1 Report
Dear authors
I can see that the manuscript is edited according to the reviewers suggestions.